# Role of Damage-Associated Molecular Patterns (DAMPS) in the Postoperative Period after Colorectal Surgery

**DOI:** 10.3390/ijms24043862

**Published:** 2023-02-15

**Authors:** María José Caballero-Herrero, Esther Jumilla, Manuel Buitrago-Ruiz, Graciela Valero-Navarro, Santiago Cuevas

**Affiliations:** 1Molecular Inflammation Group, Biomedical Research Institute of Murcia Pascual Parrilla (IMIB), 30120 Murcia, Spain; 2General and Digestive System Surgery, Morales Meseguer University Hospital, 30008 Murcia, Spain; 3Surgical Research in Health Area, Institute of Biosanitary Research Pascual Parrilla (IMIB), Department of Surgery, Pediatrics, Obstetrics and Gynecology, University of Murcia, 30100 Murcia, Spain

**Keywords:** damage-associated molecular patterns, anastomotic leak, NLRP3 inflammasome, colorectal surgery

## Abstract

Anastomotic leakage (AL) is a defect of the intestinal wall at the anastomotic site and is one of the most severe complications in colorectal surgery. Previous studies have shown that the immune system response plays a significant role in the development of AL. In recent years, DAMPs (damage-associated molecular patterns) have been identified as cellular compounds with the ability to activate the immune system. The NLRP3 inflammasome plays an important role in the inflammatory responses which are mediated by DAMPs such as ATP, HSP proteins or uric acid crystals, when found in extracellular environments. Recent publications suggest that systemic concentration of DAMPs in patients with colorectal surgery may determine the inflammatory process and have a role in the occurrence of AL and other post-surgery complications. This review provides valuable knowledge about the current evidence supporting this hypothesis and highlights the possible role of these compounds in postoperative processes, which could open a new path to explore new strategies to prevent possible post-surgical complications.

## 1. Introduction

Anastomotic leak (AL) is a potentially serious complication that occurs after intestinal surgery and is one of the most important complications in colorectal surgery both for its severity and frequency [1]. Previous studies have analyzed the value of inflammation as a predictor of the AL before clinical onset [2,3], indicating that the identification of inflammatory risk markers may be useful for early diagnosis. AL clinical onset consists of abdominal pain, whether focalized or generalized when diagnosed late. Signs of acute peritonitis could be also present with fever and high leukocytosis associated. Other symptoms such as ileus, obstruction or sepsis are less common but may indicate an even more severe leak. It is not rare that AL is actually present even in the patient remains asymptomatic for 1–2 more days, and usually the clinical features appear on the fifth or sixth day [4]. Our group has been shown that C Reactive Protein (CRP) elevation above 15Mg/d on 3º postoperative day [5,6] is an important predictor of AL, which represents a valuable tool for the early detection of AL and prevention of its harmful effects. However, the causes associated with the occurrence of AL and other postoperative complications are not yet well defined.

Previous authors claim that the occurrence of AL is the result of a complex and dynamic interaction of several factors and biological processes in which inflammation and the immune system response seem to play a significant role [5,7]. In this sense, identifying and understanding the molecular mechanisms that trigger inflammation and inflammasome activation could be decisive in designing strategies to prevent these harmful effects associated with colorectal surgery and perhaps other surgeries complications as well.

In the last decade, Polly Matzinger’s “Danger Hypothesis” [8] has emerged strongly, providing a new mechanism of activation of the innate and adaptive immune systems. The innate system is primed to sense “danger signals”, described as damage-associated molecular patterns (DAMPs), and respond to them, usually by activating the immune system and creating a pro-inflammatory environment. These DAMPs are molecules that are inherent to the organism but have a high pro-inflammatory power when they are detected in places where they are not usually present, such as in extracellular or free-form contexts, which is an indicator of tissue damage and produced in surgical processes.

These danger signals, or DAMPs, may be released by cellular and tissue rupture after surgery and could play an important role in the progression of AL creation and even in other observed post-operative complications [9]. Among the most studied DAMPs are Heat Sock Proteins (HSPs), high-mobility group protein Box 1 (HMGB1), extracellular adenosine triphosphate (ATP), uric acid crystals, free or protein-associated DNA and large concentrations of free fatty acid. These DAMPs are capable of activating the molecular platform known as the inflammasome [10], mainly in monocytes/macrophages, although also in other types of cells such as neutrophils, and even in endothelial cells [11], and rapidly induce the maturation and release into the extracellular medium of the proinflammatory cytokines IL-1β and IL-18, which is an early triggering factor of the inflammatory cascade [12,13].

This manuscript reviews the shreds of evidence published so far about the behavior of DAMPs in the appearance of AL in post-surgical processes. Due to the rupture of tissue and cells caused by surgery, a high release of DAMPs can be expected, which could trigger a sequence of inflammatory activation in the tissues, with dramatic results. This review allows us to analyze the current pieces of evidences regarding this hypothesis, in order to determine whether these DAMPs are involved in inflammasome activation, which is associated with the formation of AL and other postoperative complications, and perhaps open new prevention strategies to avoid these complications. The determination and analysis of these DAMPs may draw conclusions that help to improve our knowledge of the molecular mechanisms involved in the incidence of these pathologies.

## 2. NLRP3 Inflammasome and Inflammation

Inflammasomes have an important role in innate immunity and the initial steps for the induction of inflammatory responses. Inflammasomes are cytosolic inducible multiprotein complexes with high molecular weight [14], which are composed of an effector protein, an adaptor protein, and a sensor protein [15]. All inflammasomes known have the same adaptor and effector protein; however, different types of effectors that are activated by different signals. Three families of sensor proteins have been described and give the name to the inflammasome: (i) the nucleotide-binding domain and leucine-rich repeat receptors (NLRs), (ii) the absence in melanoma 2-like receptors (ALRs) and (iii) pyrin [16]. After the activation, some sensor proteins as NLRP3 are capable of recruiting the adaptor protein apoptosis-associated speck-like protein (ASC), which contains a caspase activation and recruitment domain (CARD) and a pyrin domain [17]. The ASC’s CARD domain can recruit the effector domain pro-caspase-1 through homotypic CARD–CARD interactions, that facilitate the autocatalytic activation of caspase-1. Caspase-1 turns pro-IL-1β and pro-IL-18 into mature pro-inflammatory cytokines [18] and activates gasdermin D (GSDMD). Gasdermin D is divided in two parts, the C-terminal fragment or GSDMD-CT, which is the repressor domain, and the N-terminal fragment or GSDMD-NT, which can oligomerize in membranes to form pores. The formation of these pores induces a type of programmed cell death called pyroptosis, that produces the liberation of cytoplasmic contents to the extracellular medium, including IL-1β and IL-18, which increase the inflammatory response [19]; see Figure 1.

To date, different inflammasomes have been described, but here we will focus on the most studied one of the NLR family, the NLRP3 inflammasome. Although inflammasomes have been mainly described in myeloid cells, NLRP3 inflammasome has also been found in other cells such as endothelial cells, hepatocytes and hepatic stellate cells (HSCs) [20]. In the canonical pathway, NLRP3 inflammasome needs two steps to become active: (1) priming and (2) activation or protein complex assembly. The first step is commonly started with the activation of a Toll-like receptor (TLR) after recognizing pathogen-associated molecular patterns (PAMPs), endogenous host-derived DAMPs, or homeostasis-altering molecular processes (HAMPs). This step increases NLRP3, pro-IL-18 and pro-IL-1β gene expression via nuclear factor kappa B (NF-κB) [12,20]. Then, a second signal is required to induce active oligomers of NLRP3 inflammasome, which can be mediated by a large variety of stimuli or DAMPs such as crystalline particles, ATP, K^+^ and Cl^−^ efflux and intracellular Ca^2+^ flux. In addition, other pathways also contribute to NLRP3 inflammasome activation, including mitochondrial dysfunction, lysosomal destabilization, metabolic alteration pathways, complement system, protein kinase R (PKR) pathway, purine receptor signaling, necroptosis, and ZBP1 pathway [21,22]; see Figure 1.

NLRP3 inflammasome activation has been shown to be highly implicated in many pathologies characterized by a predominantly pro-inflammatory environment such as cancer and autoimmunity diseases. Indeed, different kinds of NLRP3 inhibitors are being studied as anti-inflammatory therapies for some human diseases which lack effective therapies such as Alzheimer’s disease, Parkinson’s disease, multiple sclerosis, type 2 diabetes mellitus, Crohn’s disease and ulcerative colitis [23]. Most of these studies show that NLRP3 inflammasome inhibition improves some of the symptoms of these diseases. Tranilast, for example, might be used in humans as an effective and safe adjuvant therapy to relieve progressive inflammation due to severe COVID-19 [24]. Regulation of NLRP3 inflammasome by Zinc gluconate supplementation also seems to improve genital ulcer and auto-inflammatory features in Behçet’s disease patients [25].

Even though NLRP3 inflammasome overactivity may has a negative impact on many inflammatory and metabolic diseases, it plays an important and necessary role in the proper functioning of innate immunity during infectious diseases. This makes it crucial to find its right balance of activity to achieve proper cellular homeostasis and health.

## 3. Damage-Associated Molecular Patterns (DAMPs)

DAMPs are molecules found within the cell performing various functions that contribute to homeostasis [26]. Nevertheless, when these molecules are released into extracellular space, they trigger the activation of the immune system. DAMPs are released into intercellular space due to damage to the cellular structure, or cellular stress caused by various factors such as trauma, toxic substances, lack of risk and pathogens.

DAMPs can be released into the extracellular space passively, due to the death of the cell containing them (this is called passive release). However, cell death is not a completely passive process, but is regulated, or actively if molecules are released by exocytosis [26]. In a passive release, depending on the type of death the cell undergoes, different types of DAMPs are released. For example, nuclear DAMPs during apoptosis and nuclear molecules and antimicrobial enzymes during neutrophil extracellular traps (NETs) [26].

In contrast, in active release, the cell releases DAMPs by exocytosis through lysosomes and secretory exosomes. It has been observed that, in stress situations, this is the mechanism of DAMPs release [26]. These molecules are recognized through pattern recognition receptors (PRRs); mainly, these receptors recognize molecules necessary for pathogen survival, triggering the release of cytokines, chemokines and co-stimulatory molecules involved in the elimination of these pathogens [27]. However, they are also activated by DAMPs. These receptors are expressed by many cells such as phagocytes, dendritic cells and epithelial cells, both at the plasma membrane (TLR, CLR) and in the cytoplasm (ALR, RLR, NLR) [28].

If the ligand binds to TLR receptors (TLR-4), activation of the NF-𝜅β pathway occurs, inducing transcription of proinflammatory cytokines, chemokines and inflammatory mediators. A very important signalling component involved in various NF-𝜅β activation pathways, dx kinase-1, is activated by transforming growth factor β (TAK1), regulated by two subunits, TGF-Beta Activated Kinase 1 (MAP3K7) Binding Protein 1 (TAB1) and TAB2 (TAB2 binds to polyubiquitin chains, being necessary for TAK1 activation). TAK1 phosphorylates I𝜅βα activating IKK and subsequently NF-𝜅β is activated [29]. This pathway has been extensively studied in macrophages and, when activated by binding to the TLR receptor, it stimulates differentiation to M1 macrophages and stimulates the production of various proinflammatory cytokines such as IL-1, IL-6, IL-12, TNF-α, chemokines and all molecules needed for the inflammasome formation. In addition, this pathway plays an important role in T-cell differentiation and effector function [29].

Usually, activation of the adaptive immune response does not occur, because there are no T-lymphocytes or B-lymphocytes recognizing self-antigens. Otherwise, autoimmune diseases are triggered [30]; however, the activation of the innate immune system by DAMPs is referred to as a sterile immune response, an immune response that occurs upon recognition of cellular damage, without the presence of a pathogen. This recognition may be medicated by ligand recognition of NLR receptors that induces inflammasome activation. There are a variety of DAMPs that can activate the inflammasome, mainly, the inflammasome that they activate is NLRP3.

### 3.1. Types of DAMPs

DAMPs are molecules inherent to the organism, but which have a high proinflammatory power by activating the inflammasome when detected in places where they are not usually present, such as in extracellular contexts or in free form. The presence of DAMPs is an indicator of tissue damage and can be produced in surgical processes. (Table 1)

#### 3.1.1. Heat Shock Proteins (HSPs)

HSPs are a family of chaperones that maintain and minimize the damage caused by increased temperature and maintain cellular homeostasis. This type of protein is classified depending on its molecular weight in six families; in this review we are going to focus on HSP70 and HSP90.

These proteins can be released into extracellular space passively by necrosis or actively via secretory vesicles or by binding to ATP-binding cassette (ABC) transporters, which are a group of proteins that use the energy of ATP hydrolysis to transport the molecule out of the cell [31]. HSP90 proteins, together with their co-chaperone SGT1, and HSP70, have an important role in the regulation of the NLRP3 inflammasome since they stabilize the inflammasome inside the cell and keep it inactive because the HSP90 protein associates with the NACHT and LRR domains. Nevertheless, when these proteins are released into the extracellular space, they activate the innate immune system response by binding to TLRs [32,33].

The extracellular HSP70 protein induces the activation of the immune system, increasing microbicide capacity and neutrophil chemotaxis and phagocytosis. It has also been observed to stimulate tumour cell growth and resistance to apoptosis. Oppositely, the HSP90 protein activates the motility of cancer cells, their migration and subsequent metastasis. It also promotes neuronal motility and wound healing [31].

#### 3.1.2. High-Mobility Group Protein Box 1 (HMGB1)

HMGB1 is a non-histone nuclear protein that belongs to the HMGB family of proteins that binds to DNA in the nucleus and is required for transcription, gene regulation and DNA repair, and contributes to the maintenance of DNA structure [34]. The liberation of HMGB1 into extracellular space is regulated by several factors, such as oxidative stress, N-glycosylation of three residues of the protein, and ADP-ribosylation at the carboxyl-terminal end of the molecule [34].

It can be released by passive or active release. However, it is not released via the classical endoplasmic reticulum–Golgi complex pathway [26]. The most studied release models correspond to active release. One of them consists of the activation of target cells that will subsequently release HMGB1. On the other hand, the second method studied consists of the packaging of the molecule into vesicles, such as lysosomes, and its release into the extracellular space, release through lysosomes can be triggered by cytokines, signalling pathways, such as STAT or the NF-𝜅β pathway, and cell–cell interactions [34]. The HMGB1 molecule can bind to multiple receptors (RAGE, TLR4, TLR2, TLR9, etc.) causing the continuous release of cytokines and chemokines and, thus, the constant activation of the immune system [35].

#### 3.1.3. Adenosine Triphosphate (ATP)

ATP is the energy carrier molecule for all living things; however, when it is liberated into extracellular space, it acts as DAMP. ATP can be released passively in damaged tissues or cells undergoing apoptosis or pyroptosis. It can also be freed actively by two mechanisms, by exocytosis through vesicles and lysosomes or pores of the connexin and pannexin hemichannels [26]. Hemichannels open in response to various stimuli, for example, connexin is regulated by intracellular calcium concentrations, positive changes in plasma membrane voltage and phosphorylation of its serine residues, whereas exchange through pannexin is regulated by mechanical stress, increased intracellular calcium or potassium, and cleavage of the carboxyl-terminal end by caspase-3/caspase-7 and caspase-11 activated during apoptosis [26]. Binding of ATP to its receptors, P2X4R and P2X7R or P2X7, is a major stimulus for the activation of NLRP3 and NLRP1 inflammasome caspase-1. Binding of extracellular ATP to the P2X7R receptor causes its opening and the outflow of potassium into the cell. Moreover, the binding of ATP and the P2X4R receptor allows the recruitment of the pannexin hemichannel and the formation of a pore that causes the entry of extracellular material into the cell and the activation of the inflammasome. Other studies indicate that it is the high concentration of extracellular potassium that triggers inflammasome activation [32,36,37].

#### 3.1.4. Uric Acid Crystals

The presence of uric acid has been associated with an influx of water that causes the cell to swell. This hyperosmolarity is associated with an increase in macrophage size and an intracellular decrease in potassium and chloride ions which, in turn, participates in the activation of NLRP3 (not being alone sufficient for its activation) [32].

The phagolysosomes formed by macrophages upon absorbing the crystals are unstable, thus stimulating their release into the cytosol; this liberation is accompanied by cathepsin B that induces NLRP3 activation [38].

#### 3.1.5. Extracellular DNA

Extracellular DNA, mitochondrial or nuclear and free or associated with proteins, acts as DAMPs. The mechanism of release of these molecules is passively by cell death, necrosis, apoptosis, pyroptosis, etc. DNA can also be actively released through extracellular vesicles [26]. During apoptosis, mitochondrial DNA is liberated into the cytoplasm; it has been observed in those cells that mitochondria dysfunction leads to IL-1β secretion and activates the NLRP3 inflammasome [39]. In addition, there is positive feedback, when NLRP3 inflammasome activation occurs, through the outflow of intracellular potassium or the influx of extracellular calcium, plasma membrane rupture occurs and causes the release of mitochondrial DNA (mtDNA) and an increase in reactive oxygen species (ROS) that oxidize DNA. Mitochondria damage does not induce NLRP3 signalling if priming is omitted [40].

On the other hand, it has been observed that, despite the fact that NLRP3 is preferentially activated by mtDNA, AIM2, another type of is inflammasome, is activated by nuclear DNA [39]. In this case, the same happens as with NLRP3, cytoplasmic DNA interacts with ASC and activates caspase-1 to form the inflammasome [41]. Outside the cell, extracellular DNA damages neighbouring cells, mainly due to DNA bound to histones, which interacts with phosphate groups of phospholipids. In this case, NLRP3 activation occurs by inducing ROS, the outflow of potassium from the cell, and the entry of calcium into the cell. In addition, histone H4 can also activate TLR-2 and TLR-4 providing the necessary signals for inflammasome formation [41].

#### 3.1.6. Fibronectin

It has another function different from the rest. Fibronectin is a glycoprotein that stimulates the release of irisin which confers cardioprotection against various pathological stimuli. It has been studied that fibronectin type III domain 5 downregulates NLRP3, ASC, and caspase-1, one of the reasons being the reduction of ROS generation required for NLRP3 activation [42].

#### 3.1.7. Free Fatty Acids

Free fatty acids must also be considered DAMPs. It has been shown in several studies that fatty acid synthesis is involved in the activation of dendritic cells and differentiation of B lymphocytes and monocytes; it activates the immune system. Furthermore, obesity-induced danger signals, such as free fatty acids, bind to the TLR-4 receptor and induce the expression of proinflammatory cytokines and the expression of the NLRP3 inflammasome. Additionally, from free fatty acids, activation of the inflammasome occurs by the generation of mitochondrial ROS [43,44]. The imbalance between saturated and unsaturated fatty acids causes the intracellular crystallization of saturated fatty acids. On the other hand, the enzyme LDL oxidase promotes the formation of cholesterol crystals. Both crystals formed inside the cell or phagocytosed cause lysosomal destabilization, leading to the liberation of cathepsin B (involved in protein hydrolysis) which activates the NLRP3 inflammasome [44]. Inhibition of mitochondrial uncoupling protein 2 and thus fatty acid synthase enzyme (FASN), a lipogenic enzyme that promotes glucose-dependent fatty acid synthesis, was shown to suppress the expression of caspase-1, NLRP3 and pro-IL-1β [43].

A group of fatty acids that prevent inflammation is the resolvin family, which are molecules derived from the essential ω3 polyunsaturated fatty acids. It has been studied that resolvin, among others, could inhibit mechanical hypersensitivity and promote inflammatory resolution. It has been investigated that resolvin D2 (RvD2) was able to inhibit the NLRP3 inflammasome and observed that RvD2 promoted NLRP3 and IL-1β degradation through autophagy in macrophages, but it was dependent on the dose given [45,46]. Another fatty acid that also promotes the resolution of inflammation is the maresins, a family of fatty acids derived from docosahexaenoic acid. In several trials, it was observed to down-regulate the expression of cytokines, such as IL-1β, IL-18 and TNF-α. It was also seen to inhibit inflammasome activation and cell pyroptosis through the NF-𝜅β/p65 pathway [45].

**Table 1 ijms-24-03862-t001:** Types of DAMP.

Name	Origin	Cellular Function	Cellular Target That Activates the Inflammasome
Heat shock proteins (HSPs)	They are proteins that originate in stressful situations for the cell	They maintain an optimal temperature in the cell, participate in cell cycle control, and activate the immune system.	TLR4 [32,33]
High-mobility group protein Box 1 (HMGB1)	They are non-histone nuclear proteins that enable transcriptional regulation	Participate in DNA molecule stabilization, gene expression and activate the immune system	Binds to receptors such as RAGE, TLR4, TLR2, TLR9 that produce cytokines and activate the immune system [35]
ATP	These molecules are present in all cells and are indispensable for their correct functioning	It releases energy when phosphoanhydride bonds are hydrolyzed	Activates the inflammasome through the P2X4R and P2X7R receptors [32,36,37]
Uric Acid Crystals	Compound formed in the body in the catabolism of purines	Decreases nitric oxide, maintains the elasticity of blood vessels and promotes glucose absorption	TLR2 [38]
Extracellular DNA	Molecule that contains the information necessary for the development and functioning of the organism	DNA stores the information for building all cellular components	Union of ASC molecules, TLR2 and TLR4 [39,41]
Mitochondrial DNA	It is the circular chromosome found within the mitochondria	Encodes 5–10% of mitochondrial proteins	Union of ASC molecules, TLR2 and TLR4 [39,41]
Fibronectin	Glycoprotein involved in the protection of cardiac muscle	Enables the formation of fibrils in areas of tissue remodeling	TLR4 and TLR2 [42]
Free Fatty Acids	They come from triglycerides, when lipolysis of adipose tissue occurs	They act as a source of energy or precursors of other molecules	TLR4 [43,44]
Short fragments of free hyaluronan	It is a compound synthesized by the cell membrane of all cells in the organism and is located in the extracellular matrix	It is a key component for cell motility, adhesion and proliferation	TLR2 [47]

#### 3.1.8. Hyaluronic Acid

Short fragments of free hyaluronan are another DAMP that causes inflammasome activation; the production of this molecule is induced by ozone, producing airway hyperreactivity. Furthermore, hyaluronan causes cleavage and activation of pro-caspase-1, leading to activation of the inflammasome. On the other hand, hyaluronan production is related to IL-1β production and it has been shown that this hyaluronan-induced release of IL-1β is dependent on the NLRP3 inflammasome [47].

## 4. Anastomotic Leakage, Inflammation, and Fibrosis

An intestinal anastomosis is performed in the vast majority of patients who undergo a colorectal surgery that includes bowel resection. During the early postoperative days after such technique is performed, anastomotic leakage is one of the most severe complications that can lead to intensive unit care admission, reoperation, or even death. That is why studying the variables that affect AL and how to detect it in its early stages or even prevent it, is a very important topic among colorectal surgeons [2].

Inflammation is a normal response of the body to stress, cell damage, or alteration of normal homeostasis. Immune cells use several receptors to activate the immune cascade. Among these receptors, there are PRR type receptors that detect pathogen-associated molecular patterns (PAMPs), and DAMPs. These DAMPs activate the inflammasome cascade via NF-kB intranuclear signaling which upregulates the transcription of inflammasome components, mainly NLRP3. This activation increases caspase-1 cleaving of Gasdermin D and others, creating pores in the plasmatic membrane, releasing more DAMPs, and perpetuating the response [11,12,13,20].

Inflammation and inflammasome activation via NLRP3 in several organs, such as the kidney and liver, has been associated with organ fibrosis and disfunction. In the liver, cell damage releases proinflammatory molecules which activate an inflammatory cascade, leading immune cells to invade normal tissue. These cells and local Kupffer cells activate, secreting proinflammatory cytokines IL-1 and IL-18 among others, which activates hepatic stellate cells and transforms them into myofibroblasts secreting significant quantities of extracellular matrix, which may be a cause of fibrosis [20,48] Figure 2. This response, initiated via cellular damage, may also be present after major surgery. Preliminary data in mice have shown that after major surgery there is an important extracellular release of ATP, another well-known DAMP [10]. ATP activates the inflammasome cascade via P2X7R and K^+^ ionophore, acting as a second activation to NLRP3 inflammasome. Moreover, activated macrophages via inflammasome activation and proinflammatory cytokine release play an active role in fibrosis, perpetuating the inflammatory response and upregulating scar tissue formation, which if uncontrolled may cause excess fibrosis. Macrophage XBP1 mediated activation via STING cascade activates inflammasome response and produces TNF-α and TGF-β among other proinflammatory cytokines, which, when in contact with normal hepatocytes, fibroblasts, myofibroblasts or stellate cells, upregulates their extracellular matrix secretion and can lead to fibrosis [49,50]. Uncontrolled fibrosis after back surgery can lead to an excess of tissue causing radiculopathy or lumbar canal stenosis [51]. Also, after strabismus surgery, excess fibrosis can cause poor surgical results and is an important therapeutic target [52]. All this evidence leads to a possible link between inflammatory misbalance after surgery, which may cause tissue fibrosis-inducing incorrect anastomotic healing, and AL [10,12,13,20].

Furthermore, another link between AL and inflammatory response is already being used in clinical practice. CRP, an inflammation marker, is measured as a predictor of AL in these patients. An elevation during the immediate postoperative days, from the third to fifth, acts as an alert of possible complications such as AL, which is considered one of the most important complications. Several authors have demonstrated that CRP levels correlate with AL in colorectal patients [6], and it is known that NLRP3 inflammasome activation increased CRP levels [53]; however, there is no evidence about the inflammatory pathways associated with increased CRP in AL and if the DAMP release is related. Clinicians are using inflammation as a marker of complications, but this evidence points out a possible role of inflammatory modulators to prevent AL [49,54,55].

In summary, there is evidence suggesting that after surgery there is a pro-inflammatory state which could be partially mediated by DAMP secretion. Inflammation is a beneficial process that helps tissue healing, but when inflammation is uncontrolled may cause fibrosis and damage. We believe that uncontrolled inflammation and inflammasome activation can play a role in AL after surgical colonic anastomosis, so it can become a therapeutic target or a predictive tool to diagnose AL in the early stages.

## 5. Possible Role of DAMPs in Post-Surgery Complications

In the last 2 years, several studies have provided pieces of evidence about the association between DAMPs released and health complications in postoperative patients. The DAMP expression and the postoperative complication have been studied in some field of surgery in humans, and the release mtDNA have been related to some scenes like trauma, postoperative, and sepsis. A recent study in patients with cardiac bypass surgery, comparing those with minimized extracorporeal circulations vs conventional cardiopulmonary bypass, showed that the level of mtDNA was significantly higher in the group with minimized extracorporeal bypass. So, the less invasive method provides less cellular damage [56]. They also observed that patients with high mtDNA concentration (>650 copies/µL) after the beginning of the cardiac surgery, had a 5-fold higher risk for postoperative atrial fibrillation whatever the approaching group to the bypass. Patients who undergo major abdominal surgery suffer a sterile inflammatory response. *Pencovich* et al. in 2021 found that the levels of mtDNA in plasma were increased after pancreaticoduodenectomy [57]. As has been shown above, mtDNA is a known DAMP and well-known proinflammatory molecule, associated with NLRP3 activation and caspase activation, resulting in pyroptosis and cell death [10,13,20]. Some authors have also found increased expression of P2X7 after liver and skin transplantation, which is another inflammasome regulator [10]. There is also evidence that suggests that after hyperthermic intraperitoneal chemotherapy (HIPEC) and cytoreductive surgery performed in patients with peritoneal carcinomatosis, there is an increase in DAMP release, even associated with infectious complications [48].

Mitochondrial DNA gene expression was measured in 2 groups of patients, with or without multiple organ failure (MOF) after cardiac surgery. Preoperative levels were similar in both groups, however, cytochrome B gene expression and cytochrome oxidase gene expression increased after the surgery and were still high in those patients with MOF. A positive correlation (r = 0.45) was observed between sequential organ failure assessment (SOFA) score and mtDNA level in the MOF group, and mtDNA was assessed like a postoperative complications predictor [58]. In 23 patients with pancreaticoduodenectomy, mtDNA was assessed. They found that the circulating mtDNA levels were from 2-fold to 20-fold high in patients with any adverse effect during the postoperative course [57]. The levels of circulating mtDNA were also found elevated in patients after liver transplantation, compared with preoperative level and healthy controls; but this increase was bigger if an early allograft dysfunction happened [59].

HSP70 is another DAMP that has been analyzed in surgery patients. Doukas et al. in 2022 demonstrated that HSP70 level increased postoperatively in all patients after open aortic reconstruction with supra-celiac clamping, where ischemia-reperfusion phenomena are produced. They also assessed that the HSP70 level remains high in that patient with any postoperative abdominal complications such as paralytic ileus, abdominal compartment syndrome, and visceral malperfusion [60].

Xiao et al. in 2022 studied cochlear damage in mice exposed to 101 decibels of noise. The extracellular level of HMGB1 was related to noise-induced hearing loss, while those animals that were injected HMGB1-HA-tag after noise exposure reduced their hearing loss [61]. DAMPs not only have been related to postoperative complications, but they have been considered responsible for cognitive decline. The study published by Ma et al. in 2022 assessed the role of NLRP3 protein, caspase-1, IL-1β, and IL-18 levels in the risk of perioperative neurocognitive disorders after cardiac surgery in elderly patients, and concluded that high levels of NLRP3 protein just finishing the operation, were an independent predictor of neurocognitive disorder in the postoperative cardiac surgery [62].

## 6. Conclusions and Future Perspectives

Surgery is a potential event that may release an important number of DAMPs that could be involved in inflammatory responses. NLRP3 inflammasome activation is a key factor in the inflammatory response, and several studies suggest its possible role in post-surgery clinical complications. Recently, several studies highlighted the association of surgery with circulating DAMPs such as mtDNA, heat-shock protein, or HMGB1. Therefore, further longitudinal studies are needed to elucidate the effects of additional DAMPs on the inflammatory response and these effects on AL and other post-operative complications associated with surgery.

## Figures and Tables

**Figure 1 ijms-24-03862-f001:**
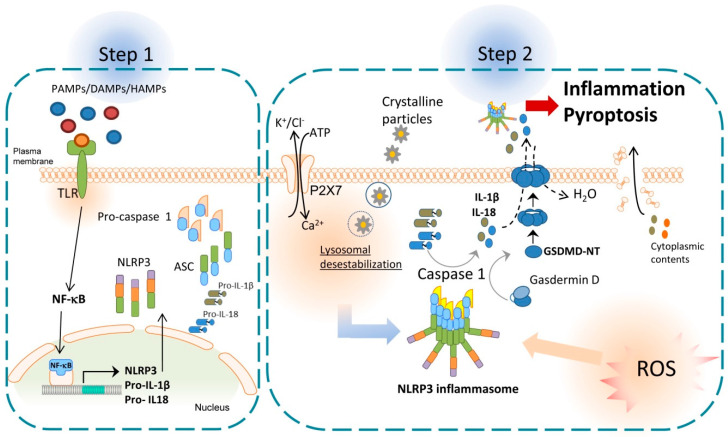
Activation of NLRP3 inflammasome. In the first step of NLRP3 inflammasome activation, TLR receptors are stimulated by pathogen-associated molecular patterns (PAMPs), endogenous host-derived damage-associated molecular patterns (DAMPs), or homeostasis-altering molecular processes (HAMPs). This activation allows the translocation of nuclear factor kappa B (NK-κB) to the cell nucleus, which, in turn, increases the transcription of the inflammasome components, pro-IL-1β and pro-IL-18. During the second step, a second signal such as P2X7 activation by extracellular ATP, lysosomal destabilization by crystalline particles or production of ROS, is required to induce the oligomerization of NLRP3 inflammasome complex which activates caspase 1. Caspase 1 turns pro-IL-1β and pro-IL-18 into mature cytokines and cleaves the N-terminus of gasdermin D which can oligomerize in membranes to form pores. Pore formation induces pyroptosis and inflammation by the liberation of cytoplasmic contents such as pro-inflammatory cytokines and NLRP3 inflammasome.

**Figure 2 ijms-24-03862-f002:**
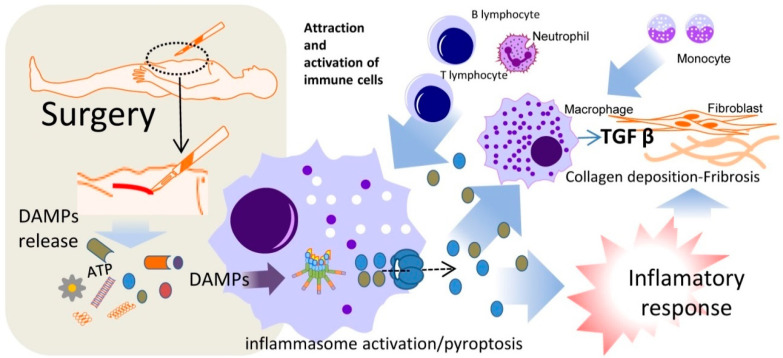
DAMP release after surgery may be involved in inflammatory response and fibrosis. Damage-Associated Molecular Patterns (DAMPs) are released in surgery into the extracellular space. These DAMPs can activate the inflammasome NLRP3 which triggers pyroptosis, the attraction and activation of immune cells, and collagen deposition and fibrosis, generating and feedback loop of inflammatory activation that could result in anastomotic leakage and other post-surgery complications.

## Data Availability

Not applicable.

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
