# Peer review of "Role of Damage-Associated Molecular Patterns (DAMPS) in the Postoperative Period after Colorectal Surgery"

_ijms, 2023, doi:10.3390/ijms24043862_

Round 1
Reviewer 1 Report
Authors have done an excellent job taking together all the current information about DAMPs and their involvement in inflammasome activation. However, it seems they are actually presenting more a kind of hypothesis on how inflammasome could be involved in the postoperative period after colorectal surgery based on other complications after major surgery (i.e. cardiac surgery, liver and skin transplantation, back surgery) instead of writing a review showing what it is recorded in literature.
Other points
- 3.1.5. First paragraph: when authors describe cells that do not possess mitochondria cannot secrete IL-1b, are they in fact referring to cells with no mitochondrial DNA?
3.1.7. Last paragraph: maresins up-regulate proinflammatory cytokines?
Author Response
We especially thank the reviewer for their comments. We have revised the manuscript in accordance with their suggestions and we believe these revisions will satisfy the issues raised by the referees. Changes in the revised manuscript are marked up using the “Track changes” function.
COMMENTS TO THE AUTHOR:
Author's Reply to the Review Report (Reviewer 1)
Authors have done an excellent job taking together all the current information about DAMPs and their involvement in inflammasome activation. However, it seems they are actually presenting more a kind of hypothesis on how inflammasome could be involved in the postoperative period after colorectal surgery based on other complications after major surgery (i.e. cardiac surgery, liver and skin transplantation, back surgery) instead of writing a review showing what it is recorded in literature.
Response: We agree with the reviewer, the main goal of this review is to put together all evidence about the possible role of DAMPs in inflammasome activation and their role in post-surgery complications, due to the emerging number of publications in the last 2 years showing evidence in this direction. However, this review not only shows a complete record in the literature about the issue, also indicates possible future directions of the future studies and speculates possible interpretations of the published data which in our opinion may add interest to the manuscript for the readers.
- 3.1.5. First paragraph: when authors describe cells that do not possess mitochondria cannot secrete IL-1b, are they in fact referring to cells with no mitochondrial DNA?
Response: We agree that this sentence was confused so it has been replaced in the manuscript for the following: …. "During apoptosis, mitochondrial DNA is liberated into the cytoplasm, it has been observed in those cells that mitochondria dysfunction leads to IL-1β secretion and activate the NLRP3 inflammasome”,,,,,
3.1.7. Last paragraph: maresins up-regulate proinflammatory cytokines?
Response: This was a mistake, maresins actually down-regulate proinflammatory cytokines, and the modification has been done in the text.

Reviewer 2 Report
|
The manuscript “Role of danger-associated molecular patterns (DAMPS) in the postoperative period after colorectal surgery” presented by María José Caballero-Herrero et al. summarizes and discusses logical links between the danger-associated molecular patterns, inflammasomes, and development and progression of the anastomotic leakage (AL). The authors compiled a sufficient amount of factual information to illustrate their main hypothesis. Unfortunately, due to early stage of investigations in this area, there is no large body of confirmed evidences on the links. However, the review can play an important transitional point in study of AL mechanisms.
|
|
Additional comments (Comments are made during continuous reading of manuscript. Therefore, answers to some raised questions may occur later in text.) |
|
Title 1) Theoretically multiple patterns should be associated with multiple roles, not a single one. However, if activation of immune response is the only role for the DAMPS it can be singular. 2) |
|
Abstract 1) Several examples of DAMPS listed in the abstract can be useful.
|
|
Introduction 1) Be more specific on “the AL before clinical onset”. What are the main characteristics of the clinical onset? 2) Sentence is too complex “Previous studies have analyzed the value of inflammation as a predictor of the AL before clinical onset [2] and the identification of risk markers for early diagnosis may minimize its severity.” 3) Typo “ Our group was been shown”. 4) Introduce a table summarizing current information on DAMPS in relation to AL and other similar conditions. The table may include information on the nature of DAMPS, their names, origins, molecular targets, and cellular and tissue effects. The first citation of such table can be done in this sentence “These DAMPs are molecules that are inherent to the organism but have a high pro-inflammatory power when they are detected in places where they are not usually present, such as in extracellular or free-form contexts, which is an indicator of tissue damage and produced in surgical processes.”. 5) Brake sentence starting “All of these are capable of activating,” into several. 6) It is important to add references for previous reviews and books on the topic allowing readers to get a more comprehensive view of the topic e.g. https://www.ncbi.nlm.nih.gov/pmc/articles/PMC7765607/ https://pubmed.ncbi.nlm.nih.gov/23075874/ 7) Theoretically, the following can be moved to the Introduction section “In summary, there is evidence suggesting that after surgery there is a pro-inflamma-tory state which could be partially mediated by DAMPs secretion. Inflammation is a beneficial process that helps tissue healing, but when inflammation is uncontrolled may cause fibrosis and damage. We believe that uncontrolled inflammation and inflammasome activation can play a role in AL after surgical colonic anastomosis, so it can become a therapeutic target or a predictive tool to diagnose AL in the early stages.” As well as some parts of section “5. Possible role of DAMPs in Post-surgery complications.”. The current introduction needs a little bit more substance. However, this is the authors’ choice. |
|
Main text 1) List the range of molecular weights for “Inflammasomes are cytosolic inducible multi-protein complexes with high molecular weight,”. 2) What are the known concentrations of the inflammasomes? 3) Be more specific in the case of “some kind of sensor proteins”. 4) Rephrase “Despite its negative impact on many inflammatory and metabolic diseases,”. 5) Make sure that the following is correct “cell releases DAMPs through lysosomes and secretory exosomes that liberate the molecules by exocytosis.” - exosomes and exocytosis. 6) Define “active transport,”. Rewrite this section to clarify types of DAMPs release and transport including “ATP-binding cassette (ABC) transporters,” and “pores in the plasmatic membrane,”. Additionally, where target cells for DAMPs should be located, and what is the relation of these sites to AL? Any spatiotemporal dynamics or dynamics in levels? 7) How HSP70 role in wound healing relate to AL? Probably, I am missing a link between HSP and AL because these molecules stabilize the inflammasome inside of the cell. 8) Typo “hypomolarity” 9) |
Author Response
Author's Reply to the Review Report (Reviewer 2)
Comments and Suggestions for Authors
The manuscript “Role of danger-associated molecular patterns (DAMPS) in the postoperative period after colorectal surgery” presented by María José Caballero-Herrero et al. summarizes and discusses logical links between the danger-associated molecular patterns, inflammasomes, and development and progression of the anastomotic leakage (AL). The authors compiled a sufficient amount of factual information to illustrate their main hypothesis. Unfortunately, due to early stage of investigations in this area, there is no large body of confirmed evidences on the links. However, the review can play an important transitional point in study of AL mechanisms.
Additional comments (Comments are made during continuous reading of manuscript. Therefore, answers to some raised questions may occur later in text.)
Title
1) Theoretically multiple patterns should be associated with multiple roles, not a single one. However, if activation of immune response is the only role for the DAMPS it can be singular.
Response: This is an interesting point, however, the immune response is activated for multiple patterns or molecular pathways, and DAMPs are only one of these pathways, indeed some of these compounds may have other actions in the cells. So, it is known that DAMPs are involved in the inflammatory response, but its action occurs in a particular homeostatic context, where other factors may be also involved in the inflammatory regulation.
Abstract
1) Several examples of DAMPS listed in the abstract can be useful.
Response: Some examples have been included in the abstract, thank you.
Introduction
1) Be more specific on “the AL before clinical onset”. What are the main characteristics of the clinical onset?
Response: AL clinical onset consists of severe abdominal pain, usually focalized on the lower half of the abdomen or generalized when diagnosed late. Signs of acute peritonitis are commonly present but not in all cases. Also, fever and high leukocytosis appear frequently. Other symptoms such as ileus, obstruction or sepsis are less common but may indicate an even more severe leak. It is not rare that Al is actually present but the patient remains asymptomatic for 1-2 more days, and usually the clinical features appear on the fifth or sixth day [1]. This information and reference have been included in introduction section of the manuscript.
2) Sentence is too complex “Previous studies have analyzed the value of inflammation as a predictor of the AL before clinical onset [2] and the identification of risk markers for early diagnosis may minimize its severity.”
Response: The sentence has been changed. …. Previous studies have analyzed the value of inflammation as a predictor of the AL before clinical onset [2], indicating that the identification of inflammatory risk markers may be useful for early diagnosis.
3) Typo “Our group was been shown”.
Response: Corrected. Thank you!
4) Introduce a table summarizing current information on DAMPS in relation to AL and other similar conditions. The table may include information on the nature of DAMPS, their names, origins, molecular targets, and cellular and tissue effects. The first citation of such table can be done in this sentence “These DAMPs are molecules that are inherent to the organism but have a high pro-inflammatory power when they are detected in places where they are not usually present, such as in extracellular or free-form contexts, which is an indicator of tissue damage and produced in surgical processes”.
Response: The table has been done and included in the manuscript.
5) Brake sentence starting “All of these are capable of activating,” into several.
Response: The sentence has been changed in the manuscript… These DAMPs are capable of activating the molecular platform known as the inflammasome…
6) It is important to add references for previous reviews and books on the topic allowing readers to get a more comprehensive view of the topic e.g. https://www.ncbi.nlm.nih.gov/pmc/articles/PMC7765607/ https://pubmed.ncbi.nlm.nih.gov/23075874/
Response: The references have been included in the manuscript. Thank you!
7) Theoretically, the following can be moved to the Introduction section “In summary, there is evidence suggesting that after surgery there is a pro-inflammatory state which could be partially mediated by DAMPs secretion. Inflammation is a beneficial process that helps tissue healing, but when inflammation is uncontrolled may cause fibrosis and damage. We believe that uncontrolled inflammation and inflammasome activation can play a role in AL after surgical colonic anastomosis, so it can become a therapeutic target or a predictive tool to diagnose AL in the early stages.” As well as some parts of section “5. Possible role of DAMPs in Post-surgery complications.”. The current introduction needs a little bit more substance. However, this is the authors’ choice.
Response: We thank the suggestion, however, this text is a summary of the section 4 and we believe that its location in the text is correct.
Main text
1) List the range of molecular weights for “Inflammasomes are cytosolic inducible multi-protein complexes with high molecular weight,”.
Response: A recent reference has been included in the manuscript which describes last information about the structure of the active NLRP3 inflammasome disc [2]. The inflammasomes is made of about eleven NLRP3 proteins (~120 kDa) and at least eleven ASCs proteins. The molecular height it may also vary depending on the type of sensor protein, and the number of ASC molecules that constitute the inflammasome which could be high and undetermined. There is an author that proposed ~700 kDa of molecular weight, but based on the last publication, we believe that the molecular weight would be bigger of 700 kDa. Therefore, due to the lack of clear information about this issue at this moment, we propose not including this information in the manuscript in order to not confuse the lector.
2) What are the known concentrations of the inflammasomes?
Response: Thank you for the suggestion, however, we consider that the presence of inflammasome is determined by the expression or the activation of the inflammasome. The concentration value is a term not appropriate to determine its presence in the cell because inflammasome is a multiprotein complex whose production varies according to the cell type, and is continually changing for numerous external and internal factors.
3) Be more specific in the case of “some kind of sensor proteins”.
Response: The information suggested for the reviewer on this point have been changes in the manuscript, you can find it in the middle of the first paragraph of point two; …“After the activation, some sensor proteins as NLRP3 are capable of recruiting the adaptor protein apoptosis-associated speck-like protein (ASC)”…
4) Rephrase “Despite its negative impact on many inflammatory and metabolic diseases,”.
Response: The change proposed for the reviewer on this point has been modified in the manuscript; you can find it in the last paragraph of point two.
5) Make sure that the following is correct “cell releases DAMPs through lysosomes and secretory exosomes that liberate the molecules by exocytosis.” - exosomes and exocytosis.
Response: The review proposed on this point have been made, you can find the correction in the manuscript at the beginning of the third paragraph of point three; …. “In contrast, in active release, the cell releases DAMPs by exocytosis through lysosomes and secretory exosomes”….
6) Define “active transport,”. Rewrite this section to clarify types of DAMPs release and transport including “ATP-binding cassette (ABC) transporters,” and “pores in the plasmatic membrane,”. Additionally, where target cells for DAMPs should be located, and what is the relation of these sites to AL? Any spatiotemporal dynamics or dynamics in levels?
Response: It was a mistake. I meant active release. The mistake has been correct in the manuscript. Thank you.
7) How HSP70 role in wound healing relate to AL? Probably, I am missing a link between HSP and AL because these molecules stabilize the inflammasome inside of the cell.
Response: These molecules stabilize the inflammasome in normal conditions, however, if there is cell damage these molecules can be released to the extracellular space and act as DAPMs activating the inflammasome and the inflammatory response.
8) Typo “hypomolarity”
Response: Corrected, thank you!
References
- Bruce, J.; Krukowski, Z. H.; Al-Khairy, G.; Russell, E. M.; Park, K. G., Systematic review of the definition and measurement of anastomotic leak after gastrointestinal surgery. The British journal of surgery 2001, 88, (9), 1157-68.
- Xiao, L.; Magupalli, V. G.; Wu, H., Cryo-EM structures of the active NLRP3 inflammasome disc. Nature 2023, 613, (7944), 595-600.
